

# Comparison of Planetary Bearing Load-Sharing Characteristics in Wind Turbine Gearboxes

Jonathan Keller[1], Yi Guo[1], Zhiwei Zhang[2], Doug Lucas[3]

[1]National Renewable Energy Laboratory, Golden CO, 80401, U.S.A.
[2]Romax Technology, Boulder CO, 80301, U.S.A.
[3]The Timken Company, North Canton OH, 44720, U.S.A.

*Correspondence to*: Jonathan Keller (jonathan.keller@nrel.gov)

**Abstract.** In this paper, the planetary load-sharing behaviour and fatigue life of different wind turbine gearboxes when subjected to rotor moments are examined. Two planetary bearing designs are compared—one design using cylindrical roller bearings with clearance and the other design using preloaded tapered roller bearings to support both the carrier and planet gears. Each design was developed and integrated into a 750-kilowatt gearbox. In field-representative dynamometer tests, the loads on each planet bearing row were measured and compared to finite element models. A significant improvement in planetary bearing load sharing was demonstrated in the gearbox with preloaded tapered roller bearings with maximum loads 20% lower than the gearbox with cylindrical roller bearings. Bearing life was calculated with a representative duty cycle measured from field tests. The predicted fatigue life of the eight-combined planet and carrier bearings for the gearbox with preloaded tapered roller bearings is 3.5 times greater than for the gearbox with cylindrical roller bearings. The influence of other factors such as carrier and planet bearing clearance, gravity, and tangential pin position error is also investigated. The combined effect of gravity and carrier bearing clearance was primarily responsible for unequal load sharing. Reducing carrier bearing clearance significantly improved load sharing, while reducing planet clearance did not. Normal tangential pin position error did not impact load sharing due to the floating sun design of this three-planet gearbox.

## 1 Introduction

Although the cost of energy from wind has declined tremendously during the past three decades (DOE, 2018), wind power plant operation and maintenance (O&M) costs are higher than anticipated and remain an appreciable contributor to the overall cost of wind energy. Wind power plant O&M averages $10 per megawatt-hour at recently installed wind plants, accounts for 20% or more of the wind power purchase agreement price, and generally increases as the wind plant ages (Wiser and Bolinger, 2017). Approximately half of the total wind plant O&M costs are related to wind turbine O&M (Lantz, 2013), and a sizeable portion of these costs is related to reliability of the wind turbine drivetrain (Kotzalas and Doll, 2010, Greco et al., 2013, and Keller et al., 2015).

Most of wind turbines installed in the United States utilize a geared drivetrain with a multi-stage gearbox including one or more planetary stages. These gearboxes must operate in a challenging, dynamic environment different from other industrial



applications (Struggl et al., 2014). In general, wind turbine gearboxes are not achieving their expected design life (Lantz, 2013), even though they commonly meet or exceed the criteria specified in standards in the gear, bearing, and wind turbine industry as well as third-party certifications. Although planet gear and bearing failures are not predominant (Sheng, 2017), they are extremely costly when they occur because they typically require replacement of the entire gearbox with a large crane.

In planetary gearboxes, equal load distribution between the planet gears is required to achieve the predicted design life. Unequal load sharing between planetary gears due to manufacturing and assembly errors has been extensively examined in the past two decades (Cooley and Parker, 2014), with analytic models often validated through finite element models or experimental measurements at fixed locations or even on rotating gearing (Mo et al., 2016, Nam et al., 2016). More recently and specifically for wind turbine gearboxes, the ability of a floating sun gear to absorb the consequences of geometrical

imperfections has been studied (Nejad et al., 2015, Iglesias et al., 2016). The effects of gravity and the drivetrain tilt angle on planet gear load-sharing and tooth-wedging behaviour were examined (Guo et al., 2014, Qiu et al., 2015). Gravity is an important factor as it introduces fundamental excitations in the rotating carrier frame of planetary gear sets. The effect of carrier bearing clearance on planetary load sharing subject to rotor moments has also been studied (Crowther et al., 2011, LaCava et al., 2013, Guo et al., 2014, Guo et al., 2015). Rotor moments impact planet load sharing, gear and bearing alignment,

and bearing contact conditions and stress (Park et al., 2013, Gould and Burris, 2016, Dabrowski and Natarajan, 2017). Rotor moments and gravity result in once-per-revolution effects in the rotating carrier frame, resulting in both increased bearing loads and fatigue and reduced bearing loads potentially causing wear or skidding conditions (Guo et al., 2014, Gould and Burris, 2016). Although it is generally agreed that a three-planet gear set with a floating central member has equal load sharing regardless of manufacturing errors (Cooley and Parker, 2014), in the wind turbine application, bearing clearance, gravity, and

rotor moments can in fact cause unequal load sharing between planet gears.

    Although load sharing between planetary gears has been examined extensively, the distribution of loads between the two or more bearing rows supporting each planet has not. In this paper, the load-sharing characteristics between the bearing rows supporting the planetary gears of two different wind turbine gearbox designs are examined and compared. This work extends previous works by the authors (LaCava et al., 2013, Guo et al., 2015, Keller et al., 2017a, Keller et al., 2017b) by examining

a wind turbine gearbox planetary section supported by preloaded tapered roller bearings (TRBs) in addition to one supported by full complement and typical caged cylindrical roller bearings (CRBs) that operate in clearance. Loads predicted by design tools are compared to test measurements across a wide range of field-representative loading conditions, and the resultant planetary section fatigue life for a duty cycle of a typical turbine is also compared. The physical phenomenon responsible for unequal load sharing of the planet bearings is identified.

## 30    2 Gearbox Design and Test Program

The National Renewable Energy Laboratory Gearbox Reliability Collaborative (GRC) has been investigating the root causes of premature wind turbine gearbox failures for over a decade. A modular 750-kilowatt wind drivetrain from a NEG Micon




750/48 wind turbine featuring a three-stage gearbox in a three-point mounted configuration, representative of most utility-scale wind turbine drivetrains, has been used for this effort, as shown in Fig. 1. In the three-point mounted configuration, the rotor and main shaft are primarily supported by a double-row spherical roller main bearing. The main shaft is connected to the planet carrier of the gearbox, which is supported by two torque arms that are mounted to the bedplate with elastomeric

5    bushings. The two torque arms, along with the main bearing, provide a total of three points of support. The three-point mounted configuration transfers torque as well as rotor moments through the gearbox, which is an important design consideration (Guo et al., 2017).

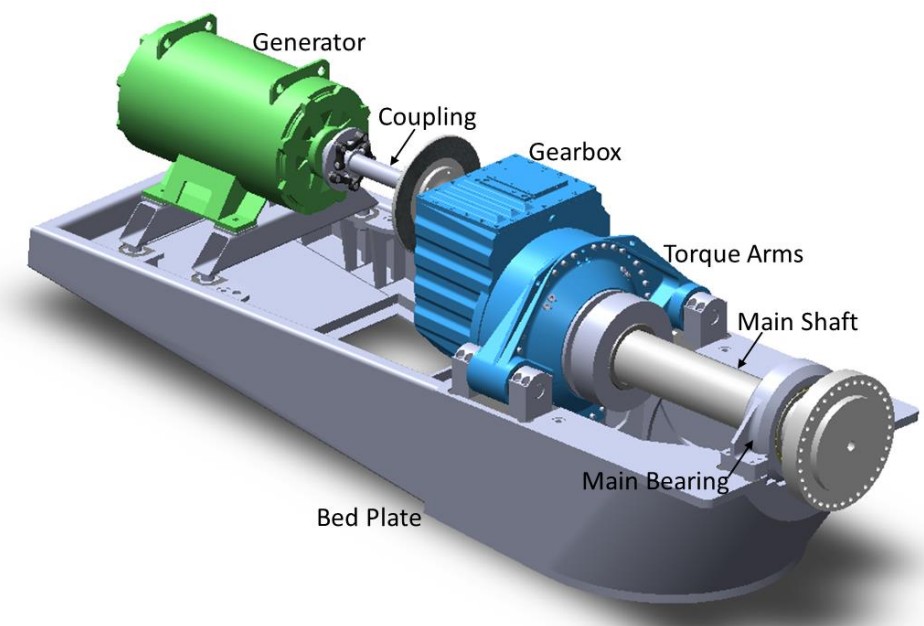

**Figure 1: Gearbox Reliability Collaborative drivetrain configuration**

10    The GRC gearbox design has a single input planetary stage followed by two parallel-shaft stages. The output shaft of the gearbox is connected to the generator with a flexible coupling. The rated rotor speed is 22.1 rpm, and with a ratio of 81.491, the gearbox increases the output speed to 1,800 rpm (Oyague, 2011, Link et al., 2011). The planetary stage features a floating sun to help equalize the load distribution among the three equally spaced planets, accomplished with a hollow low-speed shaft that has an internal spline connection to the sun pinion (Guo et al., 2013). Using this drivetrain and gearbox architecture, the

15    GRC has investigated planetary gear and bearing failure modes and load-sharing characteristics through a dedicated research and test campaign. Two different gearbox designs were purposefully developed, manufactured, and tested. As shown in Fig. 2, their primary difference is the bearing types supporting the carrier and planet bearings. One design features planet CRBs with C3 clearance and full complement carrier CRBs with CN clearance, while the other features planet and carrier TRBs under preload.



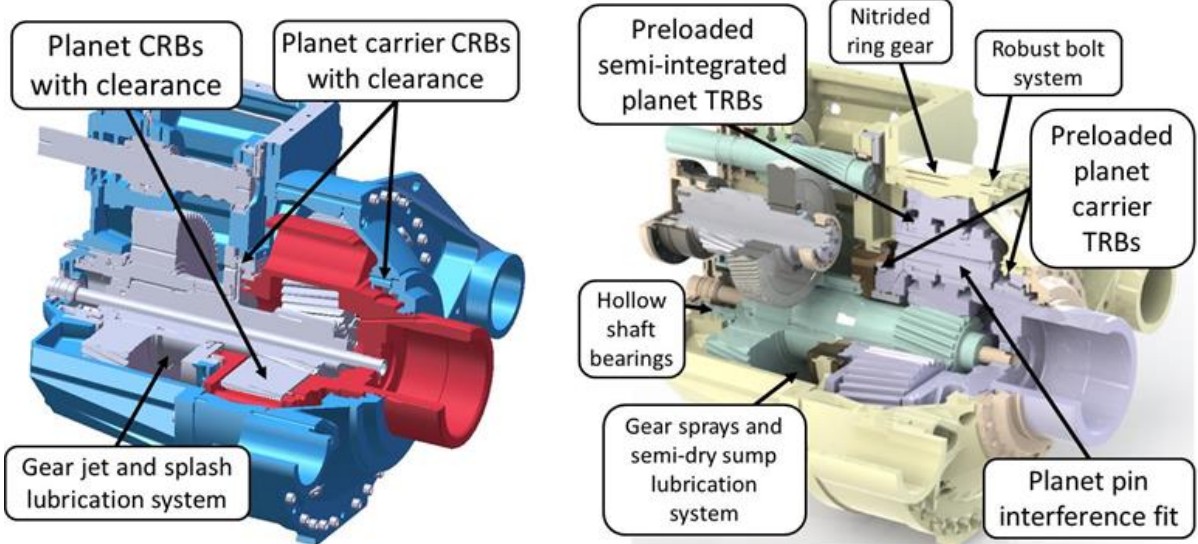

**Figure 2. Comparison of the GRC gearbox designs featuring CRBs (left) and TRBs (right). Illustration by Romax Technology (right)**

In many applications, a small preload, creating a small negative operating clearance, can optimize roller loads and maximize bearing life (Oswald et al., 2012). These preloaded bearings, along with interference-fitted planet pins, improve planet

5   alignments and load-sharing characteristics. A semi-integrated planet bearing design also increases capacity and eliminates outer race fretting. Other than these planetary system changes, including updating gear tooth microgeometry, the gearbox designs are nearly identical. The front and rear housing components, originally from a commercially available Jahnel-Kestermann PSC 1000-48/60 gearbox, and intermediate and high-speed stage gearing are used in each gearbox. Physical parameters of the planetary bearings are given in Table 1. The models used to design the gearbox with TRBs indicated that it

10  has over three times the planetary stage predicted L10 life compared to the gearbox with CRBs, like the projected increase in fatigue life in other industrial applications (Flamang and Clement, 2003, Lucas, 2005).

**Table 1. Parameters of the planetary bearings**

| Location | Cylindrical Roller Bearings | | | | Tapered Roller Bearings | | |
|---|---|---|---|---|---|---|---|
| | Designation | Bore Diameter | Original Clearance | Modified Clearance | Designation | Bore Diameter | Mounted Preload |
| Carrier bearing, rotor side | NCF1892 | 460 mm | 275 μm | 170 μm | EE244180 | 457.2 mm | 125 ± 25 μm |
| Carrier bearing, generator side | NCF1880 | 400 mm | 235 μm | 130 μm | L865547 | 381 mm | 125 ± 25 μm |
| Planet bearings, upwind and downwind | NJ2232 | 160 mm | 149 μm | 92 μm | HH231649 | 139.7 mm | 150 ± 50 μm |

In two separate test campaigns, the gearboxes were mounted in the GRC drivetrain and installed in a dynamometer at the National Wind Technology Center, as shown in Fig. 3. Steady-state, constant-speed drivetrain operations were conducted





throughout a range of power levels, from offline to the full 750-kilowatt electrical power and 325 kilonewton-meter (kNm) input torque. Representative rotor pitch and yaw moments up to ±300 kNm were applied with hydraulic actuators based on field testing (Link et al., 2011). Unique to the GRC program is that all engineering drawings, models, and resulting test data are publicly available (Keller and Wallen, 2015, Keller and Wallen, 2017).

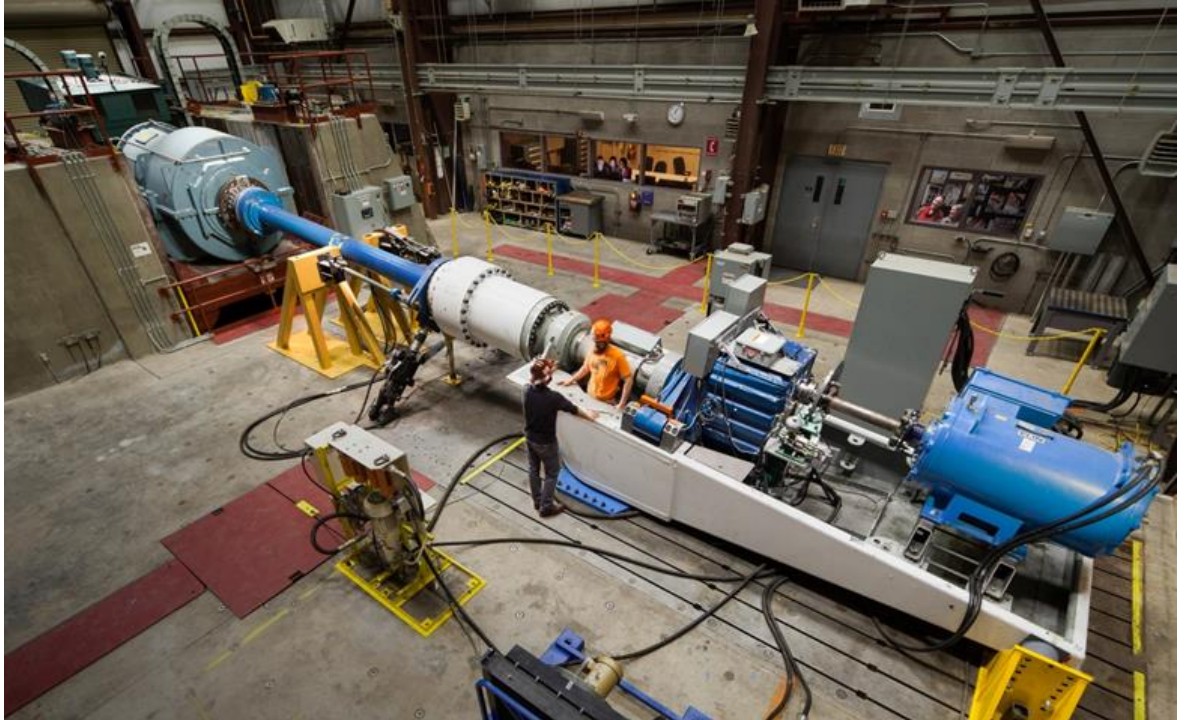

**Figure 3. Installation of the GRC drivetrain in the dynamometer. Photo by Mark McDade, NREL 32734**

Each gearbox was extensively instrumented, focusing primarily on planetary stage load-sharing characteristics. A total of 36 strain gage pair measurements were evenly placed between the upwind and downwind bearings of the three planets (A, B, and C) for each gearbox. Most of the measurements were in the expected bearing load zones, as shown in Fig. 4. The helical

10 planetary gearing causes an overturning moment on the planets, resulting in a ±20° offset of the centre of each load zone from the bearing top dead centre (TDC). The measurements were made at identical circumferential locations for the upwind and downwind bearings for the gearbox with CRBs. Two measurements were taken along the bearing inner-race width to investigate the axial load distribution between the upwind and downwind bearing rows (Link et al., 2011). Conversely for the gearbox with preloaded TRBs, the measurements focused on the circumferential load distribution with only one axial

15 measurement on each bearing inner race. Additionally, one planet (B) has measurements at 10 circumferential locations per bearing row, 9 of which span the expected load zone. The other two planets (A and C) have measurements at four circumferential locations per bearing row (Keller and Wallen, 2017). The roller load at each measurement location is determined by converting the average strain range with calibration factors determined from dedicated bench tests (van Dam, 2011, Keller and Lucas, 2017). Several thermocouples were also installed the bearing inner races for each gearbox.



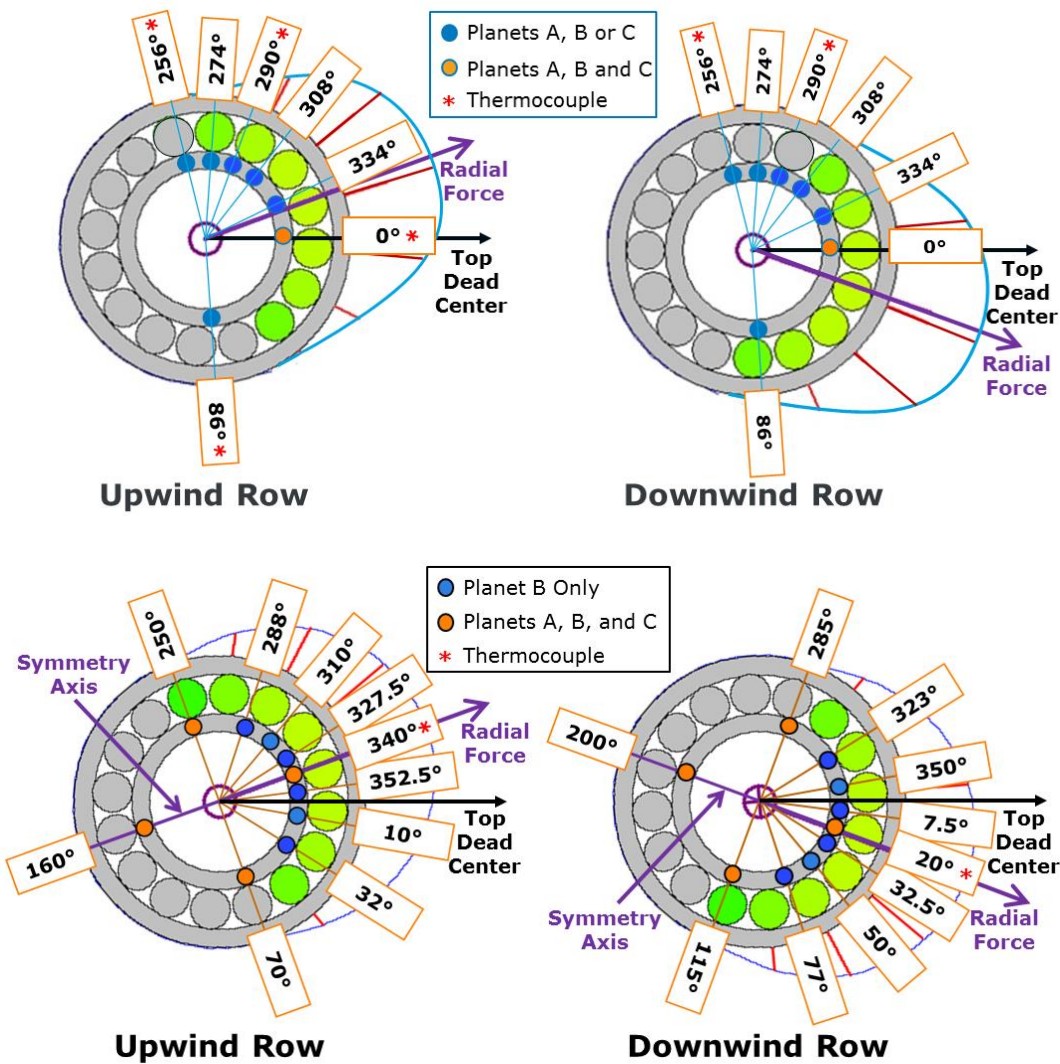

**Figure 4. Planet bearing load measurements for the gearbox with CRBs (top) and TRBs (bottom)**

## 3 Gearbox Modelling

5   The gearbox models were developed in two different finite-element, commercial software applications to predict planetary loads and load zones. The Transmission3D software application implements a three-dimensional, contact-mechanics model (Transmission3D, 2018). The entire drivetrain is represented as deformable bodies, computing gear, and bearing contacts including clearance nonlinearities with a hybrid of finite elements to predict far-field displacements and a Green's function model to predict displacements in the contact region. Known bearing clearances, preload, and pin position errors were included

10  in the model. The RomaxWind software application implements a beam-finite element representation of shafts and solid-finite element representation of the gearbox housing, gear blanks, carrier, and torque arms (RomaxWind, 2018). The gears and





bearings were modelled with semianalytical formulations that account for misalignment, area of contact under load, microgeometry, radial and axial clearances, and material properties. Static nonlinear analysis is performed for prescribed loading conditions, and the global deflections are solved simultaneously. Additionally, bearing modified L10 fatigue life calculations are made for a predetermined drivetrain torque, thrust, and rotor moment spectrum (Keller et al., 2017a).

## 4 Results and Discussion

In this section, the planetary bearing loads predicted by the model and measured in dynamometer tests are compared only for constant speed and torque, full-power cases.

### 4.1 Planet Bearing Load Zones

In this section, the bearing load zones for each gearbox are compared when the planet is at the bottom of the ring gear. The load zones for the pure torque condition are compared to those for extreme positive and negative pitch moments. As shown in Fig. 5, the upwind planet CRB load zone increases in size as the applied pitch moment increases. In general, the upwind planet bearing supports up to twice the load of the downwind bearing. The downwind planet CRB load zone is not significantly affected by the applied pitch moment. The theoretical maximum roller load (Harris and Kotzalas, 2006) of approximately 45 kilonewtons (kN) for these bearings generally correlates with the measurements and model predictions.

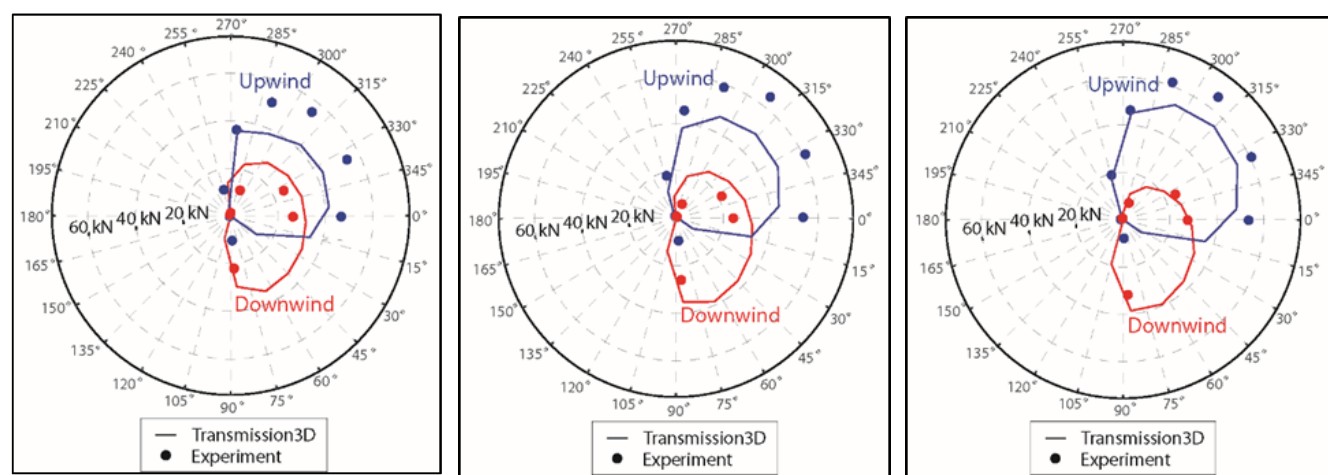

**Figure 5. Planet CRB load zones with -300 kNm (left), pure torque (middle), and +300 kNm (right) pitch moments**

In contrast, as shown in Fig. 6, the planet TRB load zones maintain their size and orientation regardless of the applied pitch moment. The more circular shape of the load zones reflects the preload in the bearings and the rigidity of the planetary system in general. The measured load zone magnitudes and orientations correlate well with the predictions, including the ±20° offset of the load zone from TDC. The theoretical maximum roller load (Harris and Kotzalas, 2006), also approximately 45 kN, again correlates with the measurements and predictions.





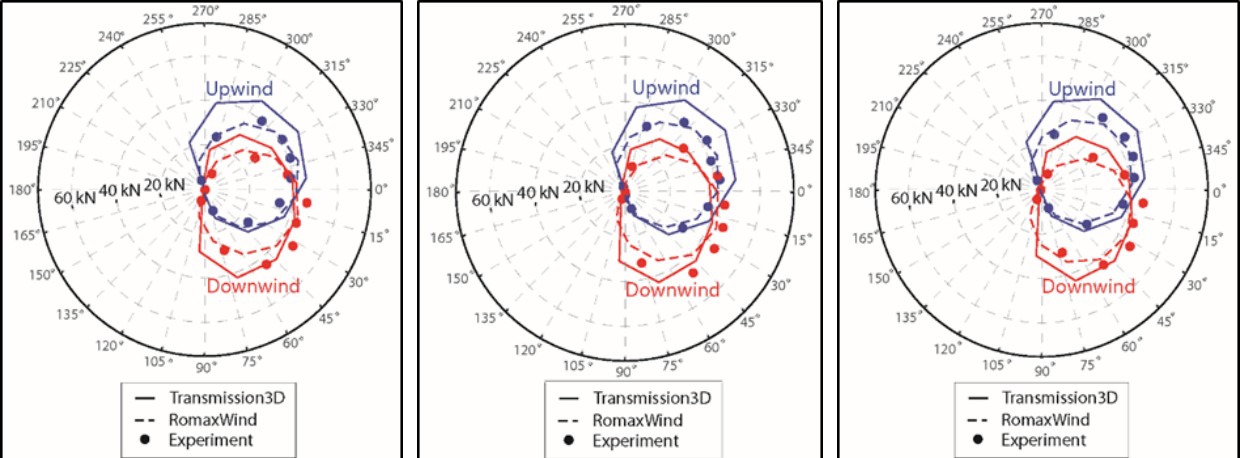

**Figure 6. Planet TRB load zones with -300 kNm (left), pure torque (middle), and +300 kNm (right) pitch moments**

## 4.2 Planet Bearing Loads

The upwind and downwind planet bearing loads can be calculated for each gearbox. For the instrumented CRBs, a direct-calibration factor is used to determine the total bearing load (van Dam, 2011, Harris and Kotzalas, 2006) from only the TDC measurement. For the instrumented TRBs, a spline fit is then used to map the entire load zone and determine the total bearing load (Keller and Lucas, 2017, Keller et al., 2017b). The total load supported by both bearings, which is the vector summation of the upwind and downwind bearing loads, can also be calculated.

Figure 7 compares the measured and predicted loads—nondimensionalized by the assumed load (i.e., one-sixth or one-third of the load at the planet centre resulting from input torque)—over a complete revolution of the planet carrier for the pure torque condition. The 0° location indicates the planet is at the top of the ring gear in its rotation, and the 180° location is at the bottom of the ring gear. The CRB loads fluctuate over the rotation and are also out of phase because of planet and carrier bearing clearances (LaCava et al, 2013). The maximum measured load carried by the upwind bearing is 1.43, or 43% more than the assumed load. The minimum measured load carried by the downwind bearing is only 0.61, or 39% less than the assumed load. In this condition, the upwind bearing is accumulating more fatigue than expected; conversely, the downwind bearing has an increased risk of skidding for a portion of the carrier rotation. Because the bearing loads are nearly 180° out of phase, there is much less fluctuation in the total load than the individual row loads. The maximum total measured bearing load is only 6% greater than assumed. The planet TRB loads are much more consistent over the carrier rotation due to the preload in the bearings and the interference-fitted planet pins. The maximum and minimum measured row loads are only 12% different than assumed, whereas the maximum measured total bearing load is only 1% more than assumed. There is good agreement between these measured loads and those predicted by Transmission3D for both gearboxes.



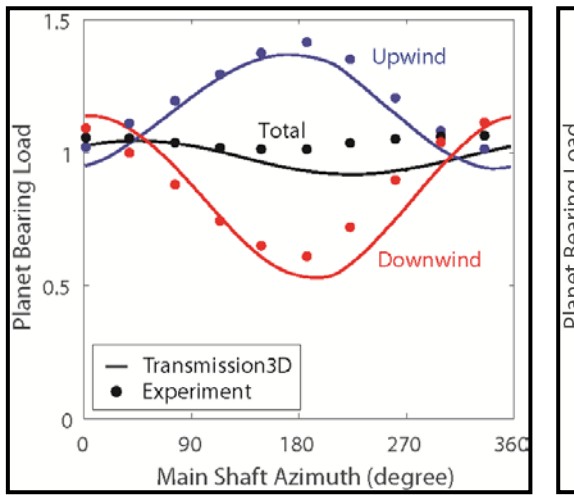
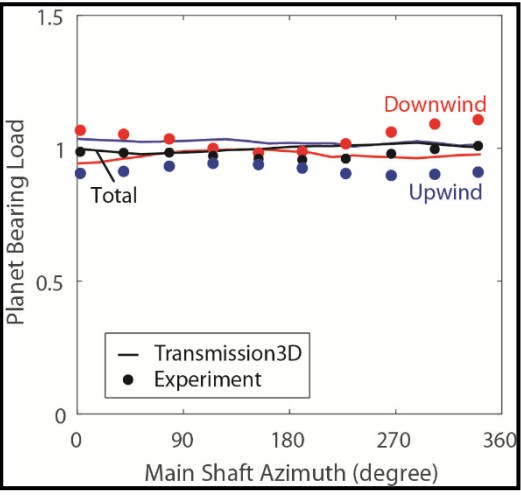

**Figure 7. Planet CRB (left) and TRB (right) loads in pure torque**

In contrast, Fig. 8 compares the same loads but with an extreme negative pitch moment. The measured upwind CRB load is relatively constant over the rotation but 25% greater than assumed. The downwind load behaviour is very similar to the pure torque condition. The net effect is that the total measured bearing load fluctuates slightly more than the pure torque condition and 15% more than assumed. The measured TRB loads again fluctuate very little—only 8% for the downwind row and 2% for the total bearing load.

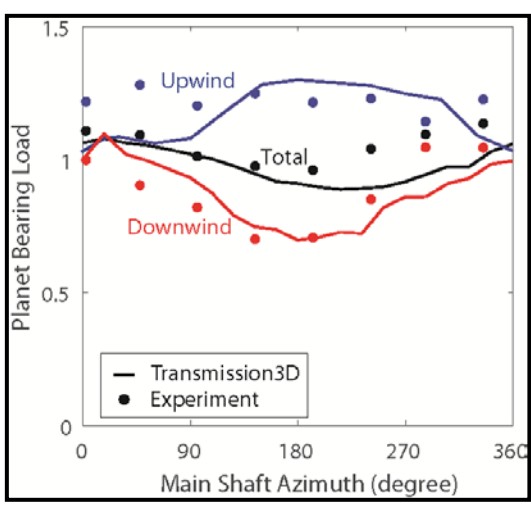
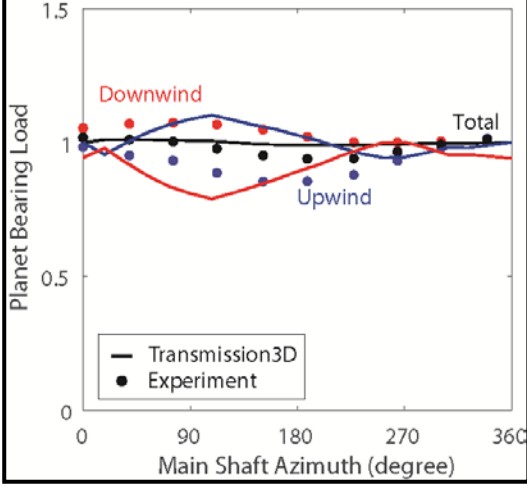

**Figure 8. Planet CRB (left) and TRB (right) loads with -300 kNm pitch moment**

Figure 9 summarizes the measured upwind and downwind planet CRB loads for all the pitch moment cases. The pitch moment changes the upwind bearing loads significantly; however, it does not affect the downwind bearing loads at all. The behaviour of the upwind bearing loads can be separated into three categories. Pure torque and positive pitch moments all essentially have the same effect, resulting in the largest variation over the carrier rotation and overall magnitude in the upwind bearing load.



Conversely, pitch moments beyond -200 kNm elevate the mean upwind bearing load with much less fluctuation over the carrier rotation. The -100 kNm pitch moment case is a transition between these two categories.

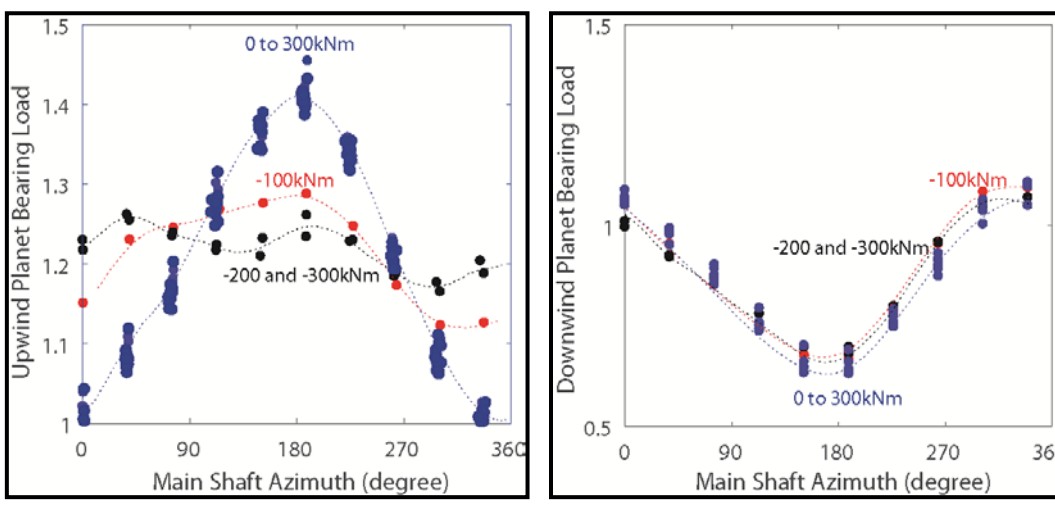

**Figure 9. Upwind- (left) and downwind-measured (right) planet CRB loads**

The bearing loads shown in these figures contain both a constant difference from the assumed load and a fluctuating component. The loads are not equally shared in practice. The constant difference is a result of deformations, displacements, and manufacturing deviations causing consistently higher loads on one planet than the others (Cooley and Parker, 2014). The fluctuating component is a result of the rotor moments and gravity, exacerbated by planet and carrier bearing clearances and resulting in misalignment in the gearbox with the CRBs, causing a once-per-revolution load variation over the carrier rotation

(Guo et al., 2015).

**4.3 Planet Bearing Load Sharing**

The accurate estimation of planet bearing loads is a crucial step in calculating the planetary load-sharing factor, also called the planetary mesh load factor ($K\gamma$). Ideally, all planets share torque equally and the planetary mesh load factor equals 1. However, because of positional-type errors and variations in tooth stiffness, International Electrotechnical Commission standard 61400-

4 assumes this factor is 1.1 for three-planet wind turbine gearboxes. In this study, the maximum load throughout the main shaft rotation, which accounts for both constant load differences and the fluctuating load from gravity and rotor moments, as shown in Fig. 7–9, is examined for comparison to this assumption.

Figure 10 compares the maximum individual bearing row load and maximum total bearing load for both gearboxes over the complete range of rotor pitch moments. The maximum total measured CRB load ranges from 1.07 in pure torque to just over

1.15 for large negative pitch moments, very close to the assumed planetary mesh load factor of 1.1. However, as shown previously, the measured CRB load carried by the upwind bearing far exceeds this, reaching 1.43 on average and as high as 1.46 in one test. Counterintuitively, this highest load occurs in the pure torque condition and is not affected by increasing the pitch moment, also as demonstrated in Fig. 9. The upwind measured CRB load does decrease with negative pitch moments;





however, it never falls below 1.26. The wide variation in the maximum CRB load can be contrasted to the consistency in the maximum TRB load. In general, the maximum TRB loads are all much closer to the assumed planetary mesh load factor of 1.1. The maximum measured downwind TRB load is 1.13 in pure torque and no more than 1.17 even for an extreme positive pitch moment—much lower than the maximum CRB load. A significant reduction of the maximum loads and improvement in

load sharing was achieved with the design changes in the gearbox with TRBs.

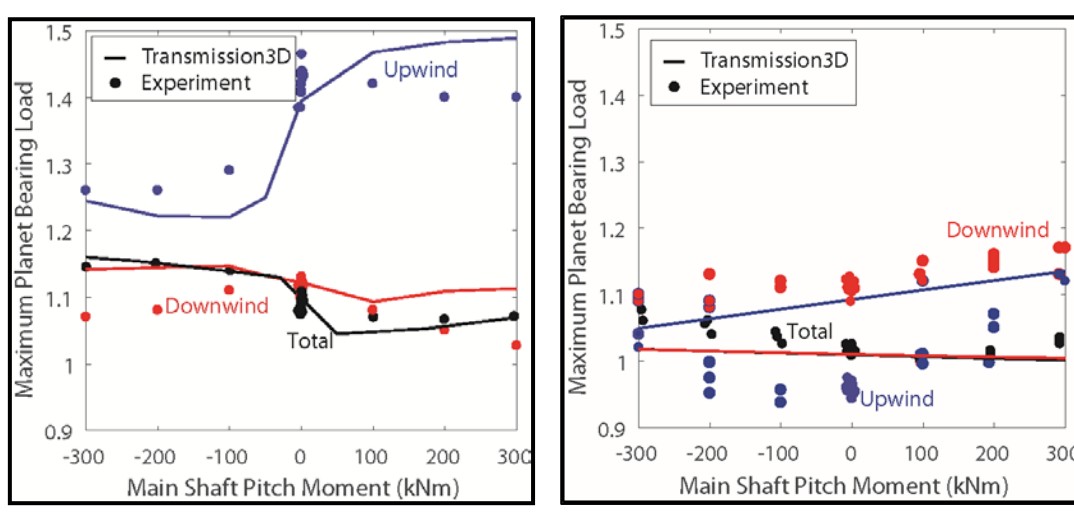

**Figure 10. Maximum planet CRB (left) and TRB (right) loads for all pitch moments**

To understand the planet bearing load-sharing behaviour, the effect of rotor pitch moment is explored on carrier bearing loads in Fig. 11. Here only the predicted loads from the model are available; measurements of the carrier bearing loads were not

acquired in tests. Carrier bearing loads are also nondimensionalized by the average of the assumed total planet bearing load. The upwind carrier CRB does not carry any load regardless of the pitch moment and neither does the downwind carrier CRB for pitch moments within ±100 kNm because of their clearances. In between ±100 kNm pitch moment, the planet CRBs are carrying all the loads. In contrast, both the upwind and downwind carrier TRBs support loads for any applied pitch moment. It is clear from Fig. 10 and Fig. 11 that for the gearbox with CRBs, rotor pitch moments can relieve the gravity load from the

main shaft and planetary system from the downwind carrier CRB and shift it to the planet CRBs. However, for the gearbox with TRBs, rotor pitch moments are essentially entirely reacted by the carrier bearings. In the three-point mount drivetrain configuration, the carrier bearing is expected to be part of the load path from the wind turbine rotor to the bed plate. In this ideal situation, the planetary gear system carries only torque and is not impacted by other loads, resulting in improved load sharing between planets and upwind and downwind rows. From this analysis, the planet carrier TRBs are carrying the moment

loads as expected, whereas the planet carrier CRBs are not.



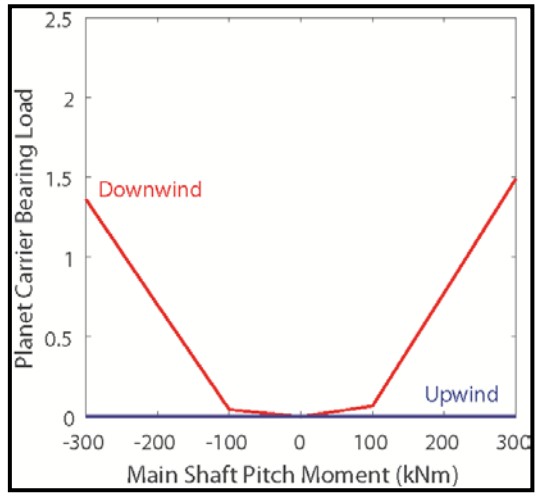
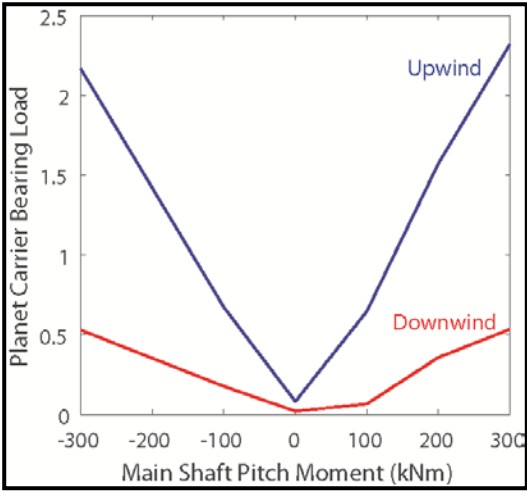

**Figure 11. Maximum carrier CRB (left) and TRB (right) loads for all pitch moments**

For comparison to rotor pitch moments, Fig. 12 shows the planet loads over the full range of rotor yaw moments for both gearboxes. The maximum upwind CRB loads occur again in pure torque conditions. Both positive and negative yaw moments

5    decrease the maximum upwind CRB load slightly; however, the measured load remains above 1.35. The total measured bearing load follows a more intuitive pattern, in which it is a minimum of 1.08 at pure torque and increases slightly to 1.13 with either positive or negative yaw moments. Rotor yaw moments have little effect on any of the TRB loads. The maximum measured load of 1.13 occurs for the downwind bearing for a positive yaw moment.

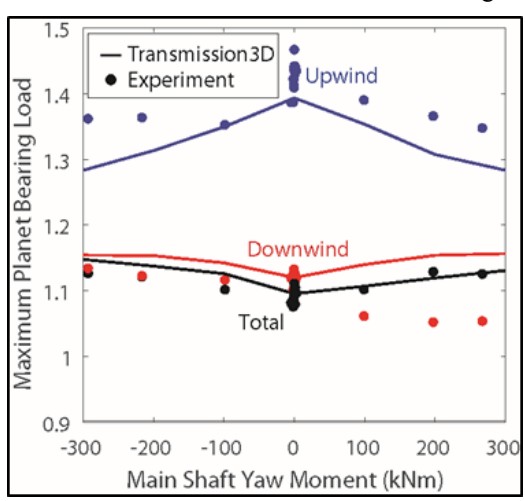
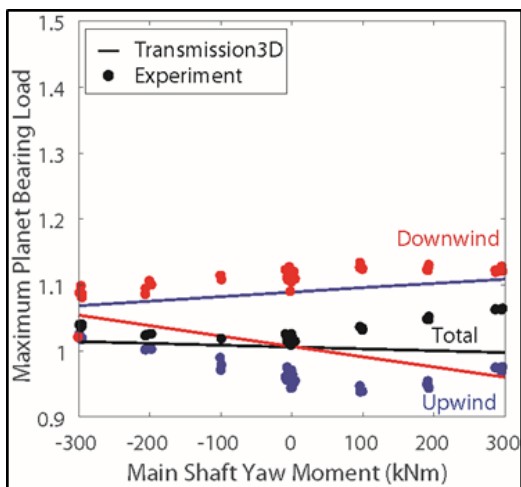

10    **Figure 12. Maximum planet CRB (left) and TRB (right) loads for all yaw moments**

Since the largest disparity in load sharing is evident even in pure torque conditions, Fig. 13 examines the maximum individual bearing row load and maximum total bearing load for both gearboxes over the complete range of pure torque conditions tested. Generally, the load increases as torque decreases – although it increases more for the CRBs than the TRBs. The maximum measured total planet bearing load increases from approximately 1.1 at full torque to 1.3 at 25 % torque for both gearboxes.




However, the measured CRB load carried by the upwind bearing increases from 1.43 on average in pure torque to as high as 1.85 at 25% torque. In contrast, the maximum measured downwind TRB load increases from 1.13 in pure torque to just 1.40 at 25% torque.

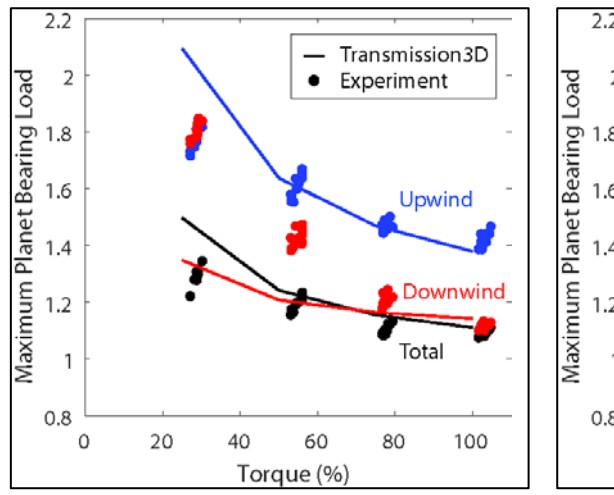
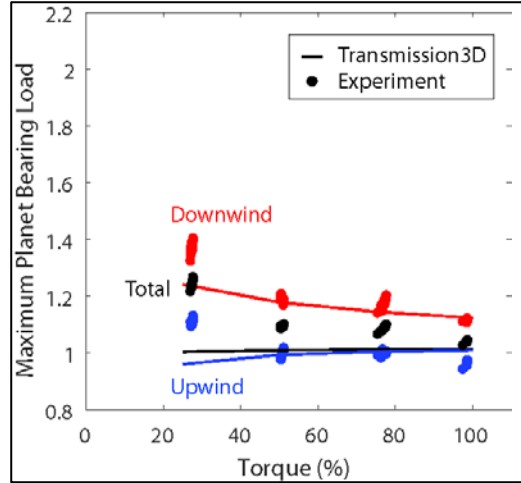

**Figure 13. Maximum planet CRB (left) and TRB (right) loads for all torque**

As shown in this section, the load sharing characteristics of the planet TRBs were significantly improved compared to the planet CRBs. However, the use of preload in these bearings does raise the question of their temperature characteristics. The measurements from the thermocouples on the bearing inner races for each gearbox, with respect to the gearbox sump temperature, are examined in Fig. 14. In this figure, the average of all thermocouples on each of the planets is examined for the full range of gearbox operating torque and applied rotor moments. The planet bearing temperatures are approximately 5 °C cooler than the sump temperatures for both gearboxes. There is little to no difference in the temperature of the planet bearing inner races between the two gearboxes. This is not necessarily a surprise as the planets are spinning at a relatively low speed compared to the bearings supporting the intermediate stage and 1,800 rpm output shaft of the gearbox. These higher speed bearings generate significantly more heat and cause the gearbox sump temperature to be higher than the planet bearing operating temperature. For reference, the absolute temperatures of the planet bearings for each gearbox were in the range of 50 to 65 °C, while the gearbox sump temperature typically ranged from 55 to 70 °C.




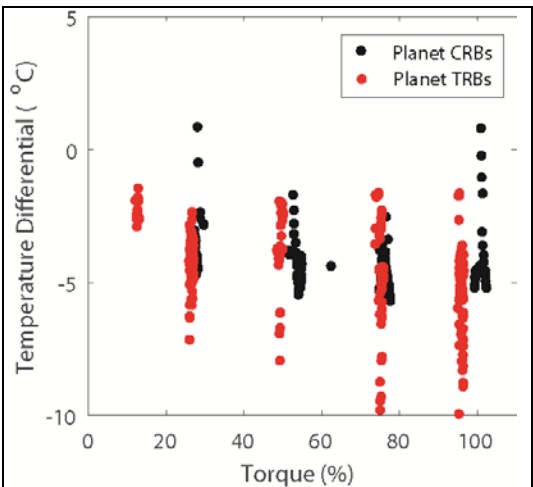

**Figure 14. Differential between the gearbox sump temperature and the average of planet bearing inner ring temperatures**

## 4.4 Planet Bearing Fatigue Life

The predicted planetary fatigue life for each gearbox was calculated using a representative drivetrain torque and rotor moment
5   spectrum derived from field measurements (Keller et al., 2017b). The modified bearing L10 life was calculated per Deutsches
Institut für Normung International Organization for Standardization 281 Beiblatt 4 (now superseded by International
Organization for Standardization technical specification 16281), including a systems life modification factor. As shown in Fig.
15, the average fatigue life of the upwind planet bearing was increased by a factor of six using the TRBs when compared to
the CRBs, in addition to a smaller life extension for both the upwind and downwind planet bearings due to the larger bearing
10   capacity in the semi-integrated design. The modified L10 life for the eight total planetary bearings (all six planet bearings and
two carrier bearings) is also shown, combined using a Weibull slope of 1.125 (Zaretsky et al., 2007). The planetary stage
bearing predicted fatigue life has been increased by a factor of 3.5 using the TRBs when compared to the CRBs. The overall
planetary bearing stage life is driven by the lowest-life components, which in this case are the planetary bearings in both
gearboxes. The carrier bearings have much longer fatigue life and thus are not shown individually.





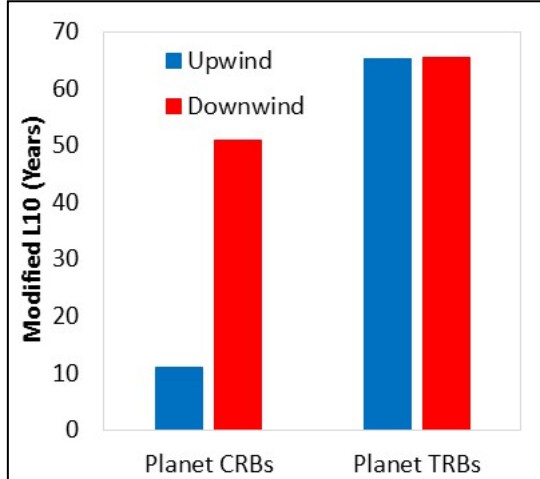
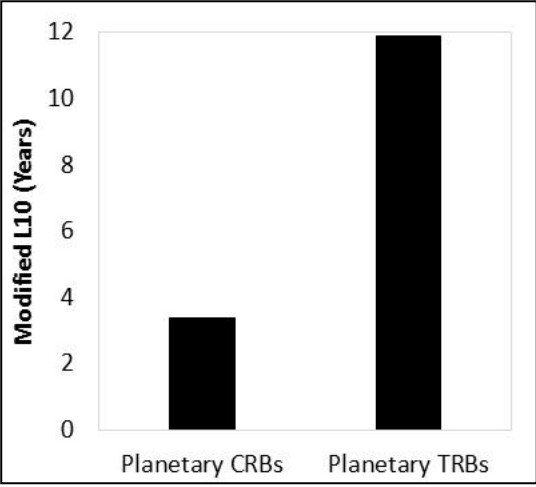

**Figure 15. Fatigue life for the planet bearings (left) and planetary bearing stage (right)**

### 4.5 Parametric Studies

The previous section examined planetary load-sharing characteristics in detail. The upwind and downwind planet bearing loads

5    were not shared equally in the gearbox with CRBs, even in pure torque conditions. In this section, the major factors responsible for the disturbed load sharing in this gearbox are examined through parametric studies of bearing clearances, gravity, and pin position error.

### 4.5.1 Effect of Bearing Clearances

The planet CRB loads were predicted with reduced clearance settings in the carrier and planet CRBs individually, as listed in

10    Table 2, for comparison to the original model clearances. As shown in Fig. 16, reducing the carrier CRB clearance from CN to C2 resulted in a noticeable improvement in load-sharing characteristics for both the upwind and downwind bearings, especially for positive pitch moments. For example, the predicted upwind CRB load decreases from 1.49 to 1.22 at +300 kNm pitch moment, a reduction of 18%. In contrast, reducing the planet CRB clearance from C3 to CN did not significantly reduce the upwind planet CRB loads for positive pitch moments, and it increased the downwind planet CRB loads.



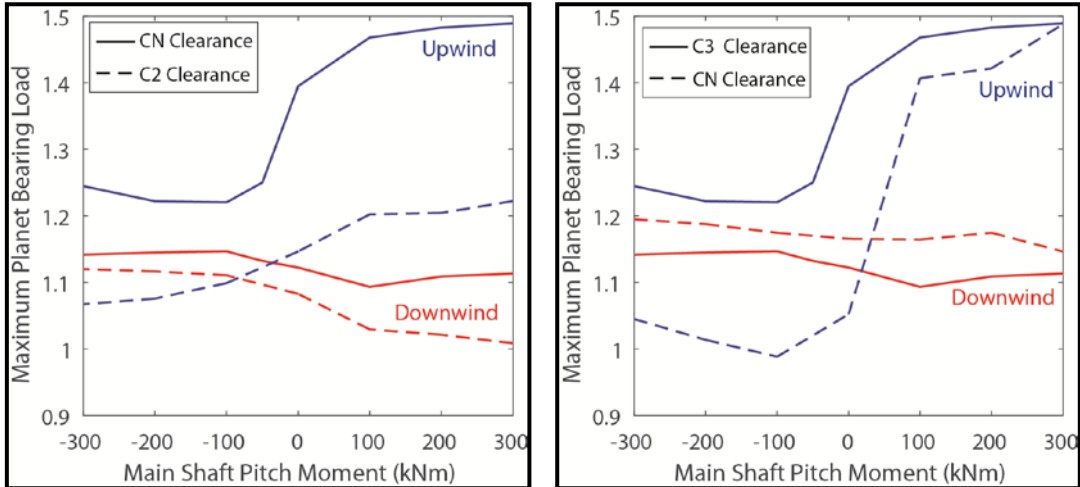

**Figure 16. Effect of carrier (left) and planet (right) bearing clearance on maximum planet CRB loads**

### 4.5.2 Effect of Pin Position Error

Tangential pin position error is one of the more common manufacturing deviations that is known to affect planetary load

5    sharing (Cooley and Parker, 2014). In this parametric study, the effect of a 15 μm tangential pin position error—a magnitude commonly considered in other applications (Singh, 2009) —on load sharing of the gearbox with CRBs is assessed. As shown in Fig. 17, the pin position error only changes the upwind planet CRB loads by less than 4% for positive pitch moments. This difference is almost negligible compared to the load fluctuations caused by other factors, which agrees with analytical results (Singh, 2009). It does not significantly change the upwind planet CRB loads in pure torque or negative pitch moments or

10    downwind CRB loads. Ideally, pin position error should not disturb load sharing with an adequately floating sun for a three-planet gearbox such as the GRC test article.

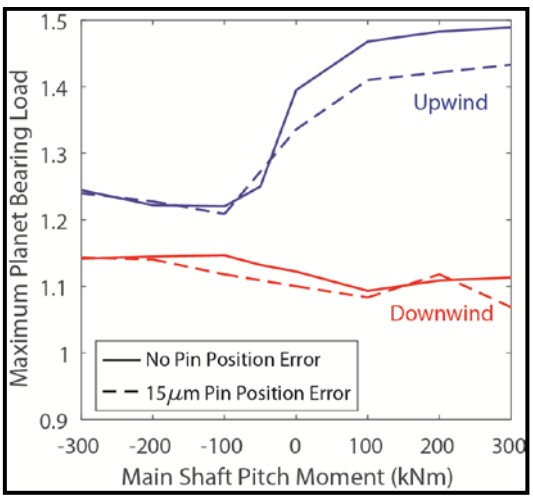

**Figure 17. Effect of tangential pin position error on maximum planet CRB loads**





### 4.5.3 Effect of Gravity

As shown previously, the interplay between the gravity load from the main shaft and planetary system and the rotor pitch moment has a significant effect on both the planet and carrier CRB loads. Figure 18 examines this further by eliminating gravity from the model. Gravity has a significant influence on the planet CRB loads, such as the effect of carrier CRB clearance.

5   Without gravity, the upwind planet CRB load reduces dramatically anywhere from 0.20 to 0.37 over the entire range of pitch moments, including a reduction from 1.40 to 1.07 at pure torque. The effect of gravity on the downwind planet CRB load is small. The effects of gravity, planetary clearances, and nontorque loads on three-point mounted wind turbine gearboxes are unavoidable and should be considered in their design. The effect of gravity on planet bearing loads can be mitigated by using carrier bearings with reduced clearances if possible.

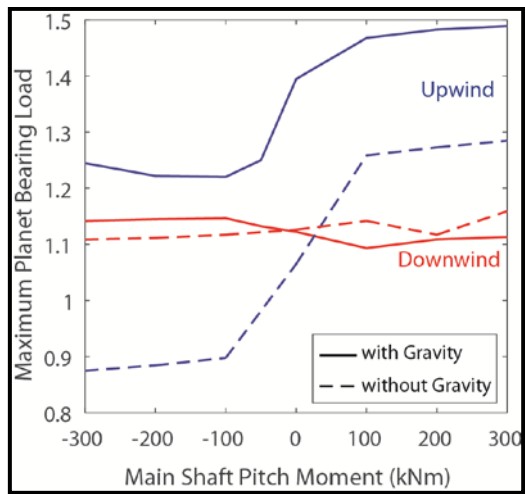

**Figure 18. Effect of gravity on maximum planet CRB loads**

### 5 Conclusions

This study compared two wind turbine gearbox planetary bearing system designs, a conventional design representing most of the gearboxes used in three-point mounted drivetrains and a new design tailored for increased planetary bearing fatigue life.

15   These two designs differ in the choice of carrier and planet bearings. The first design uses planet CRBs with clearance and the latter utilizes preloaded TRBs. Both gearboxes were designed, built, and instrumented and then tested in a dynamometer under the same set of controlled loading conditions, including rotor pitch and yaw moments. The resulting planet bearing load measurements were correlated with predictions from finite-element models of both gearboxes.

The gearbox design with preloaded TRBs demonstrated improved planetary load-sharing characteristics compared to the

20   gearbox with CRBs. The preloaded TRBs significantly reduced the planet bearing loads from a maximum of 1.46 to 1.17, a 20% reduction, in pure torque conditions. Furthermore, rotor moments did not significantly affect the upwind and downwind TRB row loads. Parametric studies indicate that the unequal load sharing in the gearbox with CRBs is primarily a result of the



combined effects of gravity, rotor moments, and bearing clearances and can be substantially improved by reducing clearance in the carrier bearings. This reduction and equalization in planet bearing loads, along with slightly larger capacity bearings through a semi-integrated design, resulted in a modified L10 life 3.5 times greater for the gearbox with preloaded TRBs than for the gearbox with CRBs.

## 5   Acknowledgements

This work was supported by the U.S. Department of Energy (DOE) under Contract No. DE-AC36-08GO28308 with the National Renewable Energy Laboratory. Funding for the work was provided by the DOE Office of Energy Efficiency and Renewable Energy, Wind Energy Technologies Office.

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
