# Peer review of "Comparison of Planetary Bearing Load-Sharing Characteristics in Wind Turbine Gearboxes"

_Wind Energy Science, 2018_

## Referee Comment (RC1) · D. Strasser (Referee) · 2 Jul 2018

D. Strasser (Referee)

dirk.strasser@zf.com

General: an articel with valuable results

Specific: p. 1, line 16: with a load dependency by the power of 3.3 for line contact rollers one would expect a live increase of about 2.1 at load reduction of 0.8

p. 2, line 5ff: one should also take into consideration the researches applied on planetary gear set load sharing back in 1990ies and 2000ff years at Ruhr University Bochum (e.g. Vriesen, Lamparski, Winkelmann, etc.)

p. 7, line 12ff: area of Downwind and Upwind is nearly the same, area can be seen as total bearing load. Further on, measured values are 20% greater than measured (upwind) and 10% smaller (downwind)

p. 7, line 21: Romax model obvously matches measurements better than Transmission 3D model

p. 8, line 13: probably it is meant: planet carrier bearing clearance leads to misalignment due to gravity force on planet carrier

p. 8, line 19: it is not clear how the interference fit influences the bearing loads. Physical effect should be described

p. 9, line 5: the individual bearing load is practical relevant, the relevance of the total measured bearing load is not clear.

p. 10, line 18f: for the sake of clarity the formulas with which the curves have been computed should be shown. It would for instance be logic to put the individual bearing loads in relation only.

p. 10, line 20f: practical experience shows significant lower values, values far above 1.1 are implausible. The physical effect should be described. At a three-planetary system the self-aligning functionality leads to the assumption that Kgamma should be nearby one.

p. 10, line 22ff: extreme values of greater than 1.2 are implausible, also values lower than 1.0

figure 11: unstetic behaviour of CRB at +/-100 kNm Needs Explanation, dito for upwind load at Zero pitch Moment. Model achitecture should be explained via sketch and text.

p. 13, line 7ff: fits to practical experience. Should be shown where t_sump is measured.

p. 16, figure 17: increased bearing load for upwind bearings at pure torque condition not plausible, physical phenomenon should be explained.

---

## Short Comment (SC1) · 10 Jul 2018

A. Natarajan

anat@dtu.dk

1) Page 2, lines 1-4; One sentence states that "wind turbine gearboxes are not achieving their expected design life" and the next sentence seems to negate this by stating "Although planet gear and bearing failures are not predominant...". Re-phrase by stating specifically what is the problem that is being addressed here.

2) Page 2, lines 15-16, "Rotor moments and gravity result in once-per-revolution effects...". Are rotor moments only 1P in frequency? Don't you have any higher frequency components from the rotor passed to the main shaft?

3)Page 5, Line 2, "Representative rotor pitch and yaw moments up to $\pm300$ kNm....". Under what situations is this a representative load? Are these for normal operating

conditions under turbulence?

4)Page 6, Line 7, "The entire drivetrain is represented as deformable bodies...". Is this really required? Provide some explanation on what role do the housing stiffness, ring gear stiffness etc. play on the loads on the bearing?

5)Page 7, Line 10, "load zones for the pure torque condition are compared to those for extreme positive and negative pitch moments". Where do you get the extreme pitch moments from? I don't think $\pm300$ kNm are extreme moments, if that is what is referred to here.

6)Page 9, Figure 7: Why should the loads on the upwind and downwind bearing be out of phase for a pure torque condition that is shown here? The explanation given is due to clearances, but possibly there is additional loading than pure torque?

7)Page 10, Figure 9, What about the main bearing load? Does the main bearing hold a part of this pitch moment load?

8)Page 11, line 11: "The upwind carrier CRB does not carry any load regardless of the pitch moment and neither does the downwind carrier CRB for pitch moments within $\pm100$ kNm because of their clearances". This is not clear.

9)Paqe15. Effect of bearing clearances: Overall all load effects shown are explained through the effect of clearances. If this is indeed the case, then the initial sections of this manuscript should explain the clearances over the different parts of the gearbox and discuss their modeling in the software.

10)Page 16, line 5: 15 $\mu$m tangential pin position error is investigated. How is this done in practice? Is this simulated or measured? Why are the load results in Fig. 17 said to negligible? They seem significant for such a small error.

11)Page 18, Line 3: The conclusions state that "resulted in a modified L10 life 3.5 times greater for the gearbox" . I don't think the cases simulated here are representative enough to compute the L10 life directly. If they are claimed to be so, then that

should be substantiated with some load simulations of normal operation. Otherwise the conclusions should just focus on the effects of clearances and gravity on loads and not extrapolate to the L10 life.

---

## Referee Comment (RC3) · H. Polinder (Referee) · 14 Jul 2018

This paper deals with modelling, design and extensive experimental model validation of gearboxes and gearbox loads. These models are then also used to calculate expected lifetime. This topic is very relevant because wind turbine gearboxes fail still fail too often, and often do not reach the design lifetime. Apparently our understanding of gearbox design is still insufficient, Therefore experimental work to check and improve models makes a lot of sense. The paper introduces its contribution in a proper way by giving an overview of previous research on this topic.

As a reviewer with a background in electrical engineering, I would like the authors to comment on the following questions:

[Figure]

1 The gearbox used in this paper was originally designed for a 750 kW wind turbine that was probably introduced about 20 years ago. Current wind turbines have torques that are more than an order of magnitude larger. What does this mean for the relevance of the paper? How do these effects scale? How has gearbox technology (gears and bearings) developed since then?

2 To me, it looks like the correlation between measurements and models is reasonable, but not excellent, while the authors characterize the correlation as good. Does it make sense to comment on the reasons for these differences? Does this mean that models should be further improved? Or does this mainly caused by manufacturing inaccuracies so that it does not make sense to try to improve models?

3 Cylindrical Roller Bearings with clearance and Tapered Roller Bearings with preload are compared, and it is concluded that TRB with preload result in a significantly longer lifetime. Does preload lead to a reduction in efficiency? Can I conclude from the temperature measurements that there is no increase in losses?

———————————————————

---

## Author Comment (AC1) · 14 Sep 2018

Thank you for your detailed comments.

Comment #1: "p. 1, line 16: with a load dependency by the power of 3.3 for line contact rollers one would expect a live increase of about 2.1 at load reduction of 0.8"

Authors' response to comment #1: The authors agree – indeed, "back of the envelope" calculations like this are very helpful. It is similar to previous work quoted in the paper of an increase of 3x in other industrial applications.

No changes were made to the manuscript.

Comment #2: "p. 2, line 5ff: one should also take into consideration the researches

applied on planetary gear set load sharing back in 1990ies and 2000ff years at Ruhr University Bochum (e.g. Vriesen, Lamparski, Winkelmann, etc.)?"

Authors' response to comment #2: The following references have been added to the paper. The authors have not yet been able to obtain English versions of 2 of the references by the deadline, so we are placing the references at the most general point in the paper.

Predki, W. and Vriesen, J. W., Calculating gear tooth corrections for planetary gears. Theoretical basis and practical benefit, Europe invites the world, International Conference on Gears, VDI reports; 1904.1; 311-326, Düsseldorf; 2005

Lamparski, C., Einfache Berechnungsgleichungen für Lastüberhöhungen in Leichtbau-Planetengetrieben, Research reports of the Ruhr University Bochum. Institute of Design Engineering, 95.3; 1-246, 1995

Winkelmann, L., Lastverteilung an Planetenradgetrieben; Schriftenreihe des Instituts für Konstruktionstechnik, Heft Nr. 87.3, Ruhr-Universität Bochum, Diss., 1987

Comment #3: "p. 7, line 12ff: area of Downwind and Upwind is nearly the same, area can be seen as total bearing load. Further on, measured values are 20% greater than measured (upwind) and 10% smaller (downwind)"

Authors' response to comment #3: Based on the experimental data (the solid circles) in Fig. 5, the area of downwind bearing is less than the upwind bearing – at +300 kNm by about half. The total load for each bearing is calculated from this area and then displayed in Fig. 7 over the entire carrier rotation. We are slightly confused by the question, as it states "measured values are 20% greater than measured (upwind) and 10% smaller (downwind)" – one of these must be predicted. At any rate, we believe the question might pertain to the fact that in Fig. 5 the area (total bearing load) calculated from the experimental data might appear to be much larger than the area (total bearing load) calculated from the Transmission 3D model. This is a reasonable assumption.

However, as stated in the manuscript "For the instrumented CRBs, a direct-calibration factor is used to determine the total bearing load (van Dam, 2011, Harris and Kotzalas, 2006) from only the TDC measurement." That is, due to the calibration test process used by van Dam, just the TDC measurement is used to directly calculate the total load – in this case there is no calculation of the area to get the total bearing load. It is an artifact of this process that the total bearing loads (experimental data and Transmission3D) shown in Fig. 7 are closer than Fig. 5 might otherwise indicate. In contrast, there is a calculation of the area (total bearing load) for the instrumented TRBs.

No changes were made to the manuscript.

Comment #4: "p. 7, line 21: Romax model obvously matches measurements better than Transmission3D model?"

Authors' response to comment #4: For the upwind bearing, the Romax results are closer to measurement. However, for downwind row bearing experimental results lie between Romax and Transmission3D modeling results. The Romax models assume rigid bearing races while Transmission3D consider flexibility of raceways. Additional sentences have been added to highlight this difference as follows:

The RomaxWind model assumes rigid bearing races while the Transmission3D includes the flexibility of the races. This results in a more circular load zone prediction for RomaxWind compared to an elliptical load zone for Transmission 3D.

Comment #5: "p. 8, line 13: probably it is meant: planet carrier bearing clearance leads to misalignment due to gravity force on planet carrier?"

Authors' response to comment #5: The authors agree that carrier bearing clearance has a greater impact on misalignment and loads than the planet bearings. This sentence has been changed slightly to read:

The CRB loads fluctuate over the rotation and are also out of phase because of the combined effect of planet and carrier bearing clearances and gravity and the resulting

gear misalignment (LaCava et al, 2013).

Comment #6: "p. 8, line 19: it is not clear how the interference fit influences the bearing loads. Physical effect should be described"

Authors' response to comment #6: Interference-fitted pins also stiffen the connection between planet pins and the carrier, reducing planet pin and thence planet gear misalignment. This is not a large effect; however, so this sentence has also been changed to:

The planet TRB loads are much more consistent over the carrier rotation due to the preload in the bearings and, to some extent, the interference-fitted planet pins that also reduce misalignment.

Comment #7: "p. 9, line 5: the individual bearing load is practical relevant, the relevance of the total measured bearing load is not clear."

Authors' response to comment #7: We agree. The individual bearing load, especially for the CRBs, is most relevant. However, the calculation and discussion of the total bearing load does still have relevancy, we think, in this paper. The main point being that the total bearing load is within the range assumed and desired in planetary gear design standards (Kgamma < 1.1) across almost all of the moments, even though the individual CRB loads are not. It is the contrast and disparity between the two that we think is interesting, especially in pure torque conditions. This discussion and comparison is included in the next section of the paper related to the planet bearing load sharing factor (Kgamma).

No changes were made to the manuscript.

Comment #8: "p. 10, line 18f: for the sake of clarity the formulas with which the curves have been computed should be shown. It would for instance be logic to put the individual bearing loads in relation only."

Authors' response to comment #8: In this section of the paper, there was no real "formula" used to translate results like Fig. 7 over a carrier revolution into the summation in Fig. 10. The paper states "In this study, the maximum load throughout the main shaft rotation shown in Fig. 7–9, which accounts for both constant load differences and the fluctuating load from gravity and rotor moments, is examined for comparison to this assumption." That is, the highest point seen in Fig. 7 is one of the points shown in Fig. 10 for pure torque. That process is repeated across all of the pitch moment cases. If this was not the point that the reviewer was commenting upon, we would ask for further clarification.

No changes were made to the manuscript.

Comment #9: "p. 10, line 20f: practical experience shows significant lower values, values far above 1.1 are implausible. The physical effect should be described. At a three-planetary system the self-aligning functionality leads to the assumption that Kgamma should be nearby one." Authors' response to comment #9: The authors agree that this is the traditional assumption and is true in many, if not most, gearbox applications. It is even a good assumption for this gearbox when examining the total bearing load. However, we find that it may not be a good assumption at all in the wind turbine gearbox application where the gearbox is mounted horizontally - especially when examining the maximum of the individual bearing load over the carrier rotation. What are thought of as implausible values, on the order of 1.4, have been demonstrated conclusively in this work by both full-scale tests and the highest fidelity finite-element models available. These values are primarily a result of the effect of gravity on the planetary system with bearing clearance; and to a certain extent also because of the effect of moments. This is exactly the main point of this paper – to demonstrate that this assumption is not true, even in what is thought of as a benign (or even best case) pure torque condition. Regarding a description of the physical effect, it is true that we have offered relatively short explanations and references to prior work at various points in the paper such as "The loads are not equally shared in practice. The constant difference is a result of deformations, displacements, and manufacturing deviations causing consistently higher

loads on one planet than the others (Cooley and Parker, 2014). The fluctuating component is a result of the rotor moments and gravity, exacerbated by planet and carrier bearing clearances and resulting in misalignment in the gearbox with the CRBs, causing a once-per-revolution load variation over the carrier rotation (Guo et al., 2015)". Having said that, these results were surprising enough to us that we have undertaken the formulation of a purely analytical description of this phenomenon as an extension to the load-sharing work of Singh (2009). For a given gearbox design (including bearing clearance), operating torque, and applied moment the load-sharing factor will be larger than 1 even for a self-aligning 3-planet design due to the effect of gravity. The effect becomes more pronounced with 4-planet and higher systems. We are preparing it in a separate manuscript, as it is generally applicable to any planetary gearbox mounted horizontally. We felt that it was beyond the scope of the present paper, as the formulation is quite lengthy. But a short explanation follows in the supplement.

Comment #10: "p. 10, line 22ff: extreme values of greater than 1.2 are implausible, also values lower than 1.0."

Authors' response to comment #10: We offer a similar response to this comment as to the comment prior. The key point is that the value that we discuss in this paper is the maximum value of the fluctuating bearing load component over the entire carrier rotation, not the average. This fluctuating effect is due to gravity and any applied moment. Although these values are not constant over the full rotation, they still have a significant impact on the predicted bearing L10 life.

No changes were made to the manuscript.

Comment #11: "figure 11: unstetic behaviour of CRB at +/-100 kNm Needs Explanation, dito for upwind load at Zero pitch Moment. Model achitecture should be explained via sketch and text.." Authors' response to comment #11: Regarding the gearbox architecture, we were hoping that Fig. 2 could suffice to show those not directly familiar with a typical 3-stage wind turbine gearbox. The carrier bearings were modeled as

springs with a constant stiffness, but with a piece-wise nonlinearity due to their individual clearances. The upwind (rotor-side) carrier CRB has larger clearance than the downwind CRB (as listed in Table 1 on Page 4). Because of this, the downwind CRB comes into contact first and reacts the applied moment and gravity loads. For this set of bearings, the upwind CRB does not carry any loads. This is certainly not desirable; the load distribution between the upwind and downwind carrier TRBs is much better. The discussion of this figure has been changed and slightly expanded to: Beyond $\pm 100$ kNm pitch moment, the downwind carrier CRB load increases while the planet CRB load does not. The downwind carrier CRB supports essentially all the additional load. Within $\pm 100$ kNm pitch moment, the planet CRBs carry any load while the carrier CRBs are both unloaded. For this gearbox, the upwind carrier CRB does not carry any load regardless of the pitch moment. This behaviour is a direct result of the relative clearances of all the carrier and planet CRBs.

Comment #12: "p. 13, line 7ff: fits to practical experience. Should be shown where t_sump is measured.."

Authors' response to comment #12: Gearbox sump temperature is measured at the bottom rear of the gearbox, near the oil return line from the sump to the oil cooler. We offer this figure below to better show the location (also provided as a reference in the manuscript), but are not sure if such a figure is worthwhile to add to the paper or what additional explanation of the location of the sump temperature measurement is worthwhile.

Comment #13: "p. 16, figure 17: increased bearing load for upwind bearings at pure torque condition not plausible, physical phenomenon should be explained."

Authors' response to comment #13: We offer the same response as comments #9 and #10. The main and most valuable conclusion from this work, we believe, is that the planet CRB loads are not equal in the wind turbine application even for a self-aligning 3-planet system in pure torque conditions. The disturbed load sharing is a direct result

of bearing clearance, gravity, and gear misalignment.

No changes were made to the manuscript.

Please also note the supplement to this comment:
https://www.wind-energ-sci-discuss.net/wes-2018-36/wes-2018-36-AC1-
supplement.pdf

---

## Author Comment (AC2) · 14 Sep 2018

The authors thank the reviewer for their comments.

Comment #1: "Page 2, lines 1-4; One sentence states that "wind turbine gearboxes are not achieving their expected design life" and the next sentence seems to negate this by stating "Although planet gear and bearing failures are not predominant...". Re-phrase by stating specifically what is the problem that is being addressed here."

Authors' response to comment #1: For clarity, the latter sentence has been changed to:

Planet bearing failures, although not the most frequent type of failure (Sheng, 2017), are extremely costly because they typically require replacement of the entire gearbox

with a large crane and thus merit investigation.

Comment #2: "Page 2, lines 15-16, "Rotor moments and gravity result in once-perrevolution effects...". Are rotor moments only 1P in frequency? Don't you have any higher frequency components from the rotor passed to the main shaft?"

Authors' response to comment #2: It is certainly true that rotor moments vary in time and in frequency. This sentence was originally intended to explain the simplest situation where even steady-state rotor moments (in the fixed frame, like those applied in dynamometer testing) and gravity (naturally constant in the fixed frame) result in 1P effects (in the carrier frame). Steady moments are the only ones that have been examined in this literature review section. For clarity, the sentence has been changed to:

Steady-state rotor moments and gravity result in a once-per-revolution variation in bearing load in the rotating carrier frame, that both increases fatigue and could cause wear or skidding (Guo et al., 2014, Gould and Burris, 2016).

Comment #3: "Page 5, Line 2, "Representative rotor pitch and yaw moments up to \_300 kNm....". Under what situations is this a representative load? Are these for normal operating conditions under turbulence?"

Authors' response to comment #3: The moments are based on a 3-month field test of this drivetrain when it was installed in a NEG Micon 750/48 turbine at a wind plant in Wyoming, USA. Moments this high were measured, but not frequently – the majority were under  $\pm$ 300 kNm. This is accounted for later in the L10 life calculations. Stating that these highest 300 kNm moments are "representative" might imply that they are common, which is not the case. This sentence has been revised to include some additional clarification and detail as follows:

Vertical and lateral forces were applied with hydraulic actuators to an adapter in front of the main bearing, resulting in bending moments up to  $\pm 300$  kNm measured on the
main shaft. This range of moments was derived from measurements on the same drivetrain when installed in a NEG Micon NM 750/48 turbine at an operational wind plant (Link et al., 2011).

Comment #4: "Page 6, Line 7, "The entire drivetrain is represented as deformable bodies...". Is this really required? Provide some explanation on what role do the housing stiffness, ring gear stiffness etc. play on the loads on the bearing?"

Authors' response to comment #4: The "entire drivetrain" being modeled with deformable bodies is a mistake that has been fixed. Other clarifications were also made, changing these sentences to:

The Transmission3D software application implements a three-dimensional, contactmechanics model (Transmission3D, 2018). The gearbox is represented with deformable bodies, including the ring gear and gearbox housing as their flexibility can affect gear misalignment and load sharing characteristics. Gear and bearing contacts, including piece-wise clearance nonlinearities, are modelled with a hybrid of finite elements to predict far-field displacements and a Green's function model to predict displacements in the contact region.

Comment #5: "Page 7, Line 10, "load zones for the pure torque condition are compared to those for extreme positive and negative pitch moments". Where do you get the extreme pitch moments from? I don't think \_300 kNm are extreme moments, if that is what is referred to here."

Authors' response to comment #5: As stated earlier, the range of main shaft moments were derived from field tests. For clarity, the sentence has been changed to:

The load zones for the pure torque condition are compared to those for the highest pitch moments.

Comment #6: "Page 9, Figure 7: Why should the loads on the upwind and downwind bearing be out of phase for a pure torque condition that is shown here? The explanation

WESD
given is due to clearances, but possibly there is additional loading than pure torque?"

Authors' response to comment #6: The gravity load is the additional load. The difference in phase between the upwind and the downwind loads is a direct result of the difference in size of the load zones in Fig. 5. At the 180 degree location, the larger upwind load zone results in a higher load while the smaller downwind load zone results in a lower load. The authors are actually developing a separate journal manuscript which examines this behavior analytically, but for now just refer to previous modeling studies. For additional clarification, the sentence has been changed to:

The CRB loads fluctuate over the rotation and are also out of phase because of the combined effect of planet and carrier bearing clearances and gravity and the resulting gear misalignment (LaCava et al, 2013).

Comment #7: "Page 10, Figure 9, What about the main bearing load? Does the main bearing hold a part of this pitch moment load?"

Authors' response to comment #7: Rotor moments can certainly be carried by both the main bearing and low-speed stage of the gearbox, depending on their relative clearances and stiffnesses. As mentioned in the response to comment #3, the moments discussed herein were actually measured on the main shaft and so are related but not equal to the rotor moments. The authors have made minor changes in the manuscript text to distinguish between the rotor moment itself, which then results in the main shaft pitch and yaw moments that were used during the tests. Note that the figures in question use this easily measurable quantity, "Main Shaft Pitch Moment" and "Main Shaft Yaw Moment", on the x-axis. The main spherical bearing in this case most likely supports a minor and unknown moment, but this uncertainty is not relevant as all moments were measured on the main shaft – including the field tests.

No changes were made to the manuscript.

Comment #8: "Page 11, line 11: "The upwind carrier CRB does not carry any load
regardless of the pitch moment and neither does the downwind carrier CRB for pitch moments within \_100 kNm because of their clearances". This is not clear."

Authors' response to comment #8: The authors agree that this sentence was confusing. The section has been modified and also better linked to the previous result:

To better understand the planet bearing load-sharing behaviour shown in Fig. 10, the effect of pitch moments on carrier bearing loads is explored in Fig. 11. Here only the predicted loads from the model are available; measurements of the carrier bearing loads were not acquired in tests. Carrier bearing loads are also nondimensionalized by the average of the assumed total planet bearing load. Beyond  $\pm 100$  kNm pitch moment, the downwind carrier CRB load increases while the planet CRB load does not. The downwind carrier CRB supports essentially all the additional load. Within  $\pm 100$  kNm pitch moment, the planet CRBs carry any load while the carrier CRBs are both unloaded. For this gearbox, the upwind carrier CRB does not carry any load regardless of the pitch moment. This behaviour is a direct result of the relative clearances of all the carrier and planet CRBs.

Comment #9: "Paqe15. Effect of bearing clearances: Overall all load effects shown are explained through the effect of clearances. If this is indeed the case, then the initial sections of this manuscript should explain the clearances over the different parts of the gearbox and discuss their modeling in the software."

Authors' response to comment #9: The planetary bearing types and clearances were described earlier in the paper in Table 1. The description of the "piece-wise" CRB clearance modeling has been added to the modeling section as mentioned in the response to comment #4.

Comment #10: "Page 16, line 5: 15 ïAmm tangential pin position error is investigated. How is this done in practice? Is this simulated or measured? Why are the load results in Fig. 17 said to negligible? They seem significant for such a small error."
Authors' response to comment #10: Pin position error was simulated in the model. There is no direct measurement of the error itself. Pin position error commonly exists during the manufacturing process, as stated in the introduction. The authors agree that describing the effect of pin position error as "negligible" is an overstatement, so this sentence has been changed to:

This effect is much smaller than the load fluctuations caused by other factors,....

Comment #11: "Page 18, Line 3: The conclusions state that "resulted in a modified L10 life 3.5 times greater for the gearbox" . I don't think the cases simulated here are representative enough to compute the L10 life directly. If they are claimed to be so, then that should be substantiated with some load simulations of normal operation. Otherwise the conclusions should just focus on the effects of clearances and gravity on loads and not extrapolate to the L10 life."

Authors' response to comment #11: As mentioned earlier, the loading conditions were measured in field tests over a 3-month field test. Although brief, this test period was used to calculate a representative duty cycle for the turbine. It was this duty cycle that was then used to make the L10 life calculations. The life calculations are based on this whole duty cycle, not just the extreme load cases presented here. The duty cycle is actually comprised of mostly low or near pure-torque conditions, and indeed the gearbox with CRBs shows disturbed load-sharing even in what is thought of as this "benign" (or even best case) condition. Regardless of how accurate the duty cycle may be, what is important to quantify is the relative difference in L10 life between the two gearboxes (which is significant) rather than just stopping at an examination of the difference in planet bearing loads.

No changes were made to the manuscript

Please also note the supplement to this comment: https://www.wind-energ-sci-discuss.net/wes-2018-36/wes-2018-36-AC2-

---

## Author Comment (AC3) · 14 Sep 2018

Comment #1: "The gearbox used in this paper was originally designed for a 750 kW wind turbine that was probably introduced about 20 years ago. Current wind turbines have torques that are more than an order of magnitude larger. What does this mean for the relevance of the paper? How do these effects scale? How has gearbox technology (gears and bearings) developed since then?"

Authors' response to comment #1: This historical perspective and future outlook is certainly true. However, for many modern turbines the drivetrain architecture (3-point) and gearbox planetary design (3-planets supported by CRBs and a floating sun) have not changed and are still commonly used. For larger turbines, the torque has increased

but so has the rotor moments and size (mass) of the gearbox. It is anticipated that the planetary load sharing problem persists for modern, large wind turbine gearboxes. The authors are actually developing a separate journal manuscript which examines this behavior analytically – the formulation itself being worthy of a journal manuscript in our estimation. For a given torque, rotor moment and mass for a larger drivetrain this question could be examined in more detail. But, we felt that it was beyond the scope of the present paper.

No changes were made to the manuscript.

Comment #2: "To me, it looks like the correlation between measurements and models is reasonable, but not excellent, while the authors characterize the correlation as good. Does it make sense to comment on the reasons for these differences? Does this mean that models should be further improved? Or does this mainly caused by manufacturing inaccuracies so that it does not make sense to try to improve models?"

Authors' response to comment #2: This is a valid point and question, and was part of the reason the authors undertook the parametric studies – especially for pin position error. In retrospect, for the gearbox with CRBs we wish we had acquired more data points (instead of just 1) for the pitch and yaw moment cases like we did for pure torque conditions (this can be seen, for example, in Fig. 10). This would allow a better assessment of how well the models and experiments match. We did acquire more of these points for the gearbox with TRBs and we think it improves the quality of the work. The Transmission3D software is the highest-fidelity software that the authors are aware of for this type of gearbox modeling.

No changes were made to the manuscript.

Comment #3: "Cylindrical Roller Bearings with clearance and Tapered Roller Bearings with preload are compared, and it is concluded that TRB with preload result in a significantly longer lifetime. Does preload lead to a reduction in efficiency? Can I conclude from the temperature measurements that there is no increase in losses?"

Authors' response to comment #3: Also a valid concern, and it was the motivation for examining the temperature measurements as you have surmised. To better address this fact, the authors have added a phrase stating:

There is little to no difference in the temperature of the planet bearing inner races between the two gearboxes and thus most likely little to no impact on the gearbox efficiency.

Please also note the supplement to this comment:
https://www.wind-energ-sci-discuss.net/wes-2018-36/wes-2018-36-AC3-supplement.pdf